# Tetrel Bonds between Phenyltrifluorosilane and Dimethyl Sulfoxide: Influence of Basis Sets, Substitution and Competition

**DOI:** 10.3390/molecules26237231

**Published:** 2021-11-29

**Authors:** Xiulin An, Xin Yang, Qingzhong Li

**Affiliations:** 1College of Life Science, Yantai University, Yantai 264005, China; anxiulinli@sina.com; 2The Laboratory of Theoretical and Computational Chemistry, School of Chemistry and Chemical Engineering, Yantai University, Yantai 264005, China; yangx@ytu.edu.cn

**Keywords:** tetrel bonds, phenyltrifluorosilane, dimethyl sulfoxide, substituents, competition

## Abstract

Ab initio calculations have been performed for the complexes of DMSO and phenyltrifluorosilane (PTS) and its derivatives with a substituent of NH_3_, OCH_3_, CH_3_, OH, F, CHO, CN, NO_2_, and SO_3_H. It is necessary to use sufficiently flexible basis sets, such as aug’-cc-pVTZ, to get reliable results for the Si···O tetrel bonds. The tetrel bond in these complexes has been characterized in views of geometries, interaction energies, orbital interactions and topological parameters. The electron-donating group in PTS weakens this interaction and the electron-withdrawing group prominently strengthens it to the point where it exceeds that of the majority of hydrogen bonds. The largest interaction energy occurs in the *p*-HO_3_S-PhSiF_3_···DMSO complex, amounting to −122 kJ/mol. The strong Si···O tetrel bond depends to a large extent on the charge transfer from the O lone pair into the empty *p* orbital of Si, although it has a dominant electrostatic character. For the PTS derivatives of NH_2_, OH, CHO and NO_2_, the hydrogen bonded complex is favorable to the tetrel bonded complex for the NH_2_ and OH derivatives, while the σ-hole interaction prefers the π-hole interaction for the CHO and NO_2_ derivatives.

## 1. Introduction

Recently, the tetrel bond has attracted more attention due to its potential applications in crystal materials [1,2,3], chemical reactions [4,5,6], and biological systems [7]. It is defined as an attractive interaction between a Group 14 atom and a Lewis base [1]. Like hydrogen bonds, the Lewis bases in tetrel bonds are usually nitrogen- and oxygen-containing molecules, though other types of Lewis bases are also found [8,9,10,11]. For example, this attractive interaction has been reported in some simple complexes of SiF_4_···NH_3_ [12,13,14] and SiF_4_···H_2_O [15]. However, this interaction was named a tetrel bond until 2013 [1]. Actually, the attractive interaction between a Group 14 atom and a Lewis base has been understood with molecular electrostatic potentials (MEPs), and it was found that there is a region with positive MEPs (σ-hole) at the outer side of the tetrel atom [16]. Now, the tetrel bond has been accepted as a σ-hole interaction by many researchers. In most cases, the σ-hole on the tetrel atom enlarges in the order C << Si < Ge < Sn < Pb due to the larger polarization and smaller electronegativity of the heavier tetrel atom [16].

The strength of tetrel bond is related to not only the nature of the tetrel atom but its substituents adjoined to it as well. Specifically, the electron-withdrawing group in the tetrel donor enlarges the σ-hole on the tetrel atom and thus strengthens tetrel bond. For example, the interaction energy is −2.6 and −10.5 kcal/mol in the complexes of SiH_4_···NH_3_ and SiF_4_···NH_3_, respectively [12]. The strength of the tetrel bond can also be regulated by the cooperative effect between the tetrel bond and other interactions [17,18,19,20,21,22,23,24,25,26,27,28,29,30]. Such a cooperative effect is important in constructing crystal materials and maintaining the conformation of macromolecules. In our paper, we studied the modulation of protonation on the interaction mode and strength between pyridine-TF_3_ (T = tetrel) and NH_3_ [31]. A F∙∙∙H hydrogen bond in the neutral complex pyridine-CF_3_···NH_3_ is changed to be a tetrel bond in the protonated analogue. The protonation has a prominent enhancing effect on the strength of tetrel bonding, with an increase of interactions energy from −14 to −30 kcal/mol.

A focus was also paid on the competition between the tetrel bond and other interactions [9,17,32,33,34]. In NCCH_3_···CH_3_ and CNCH_3_···CH_3_ [9], the single-electron tetrel bond is formed between NC/CN and the methyl radical; on the other hand, a C–H···N/C hydrogen bond is also present. Both interactions have comparable interaction energies, thus there is competition in the formation of both hydrogen-bonded and single-electron tetrel-bonded complexes. F_2_CX (X = Se and Te) can form a tetrel bond through the π-hole on the carbon atom and a chalcogen bond through the σ-hole on the chalcogen atom, respectively [17]. The results showed that F_2_CSe forms a stronger tetrel bond and F_2_CTe engages in a stronger chalcogen bond. Even so, there is competition between tetrel and chalcogen bonds. In complexes of DMSO and TF_3_X (T = C and Si; X = halogen) [32], CF_3_X is favorable to form the halogen-bonded complexes and SiF_3_X prefers the formation of the tetrel-bonded complexes. The halogen bond can compete with the tetrel bond in the complexes of CF_3_X, whereas the tetrel bond dominates over the halogen bond in the complexes of SiF_3_X. When NH_3_ approaches XCN (X = F, Cl, Br, I) along its molecular axis or a perpendicular π-hole on the carbon atom, a σ-hole halogen bond and a π-hole tetrel bond are formed, respectively [33]. Moreover, the latter geometry is favored for X = F and the σ-hole structure is preferred for the heavier halogens.

In the present paper, we study the tetrel bonded complex between phenyltrifluorosilane (PTS) and dimethyl sulfoxide (DMSO). PTS is an original and effective reagent and synthon in organoelemental and organic synthesis [35,36]. DMSO is a good solvent in chemical reactions and biological proceeds [37]. Thus, it is important to study the interaction between both molecules. First, a reliable basis set is considered for studying the tetrel bond between them. Then this tetrel bond is characterized in views of geometries, energies, orbital interactions and topological parameters. Thirdly, the substitution effect on the strength of the tetrel bond is investigated in the para-derivatives of PTS. Finally, the competition between the tetrel bond and other interactions is studied.

## 2. Theoretical Methods

The geometries of complexes and monomers were optimized at the second-order Møller−Plesset perturbation theory (MP2) level of theory with three basis sets including 6-311++G(d,p), aug-cc-pVDZ, and aug’-cc-pVTZ, which is the Dunning aug-cc-pVTZ basis set with diffuse functions removed from H atoms. Frequencies were computed with the former two basis sets to identify local minima on the surfaces. Optimization and frequency calculations were carried out using the Gaussian09 program [38].

Interaction energies were computed as the difference between the energy of complex and the sum of energy of monomer in the complex. Interaction energies were obtained with the above three basis sets and with the aug-cc-pVTZ and 6-311++G(3df,2p) basis sets on the MP2/6-311++G(d,p) geometries. Interaction energies were then corrected for the basis set superposition error (BSSE) by using the Boys−Bernardi counterpoise technique [39]. The similar theoretical methodology was also performed for other types of noncovalent interactions [40,41,42,43,44].

Molecular electrostatic potentials were derived via the wavefunction analysis-surface analysis suite (WFA-SAS) program [45] using the MP2/aug’-cc-pVTZ electron density. The Natural Bond Orbital (NBO) method [46] has been used to analyze the orbital interactions and charge transfer. Since MP2 orbitals are nonexistent, the second-order perturbation energies were evaluated using WB97XD method with the aug’-cc-pVTZ basis set on the MP2/aug’-cc-pVTZ geometries. The “atoms in molecules” (AIM) theory of Bader [47] was applied, and the bond critical points were analyzed in terms of electron densities, Laplacians and energy densities. The AIM calculations were performed with the use of the AIM2000 program [48].

## 3. Results and Discussion

### 3.1. Selection of Basis Sets

Figure 1 shows the scheme of ten complexes studied here. Table 1 presents the interaction energies of these complexes at the MP2 level with different basis sets including aug-cc-pVDZ, aug-cc-pVTZ, aug’-cc-pVTZ, 6-311++G(d,p) and 6-311++G(3df,2p). Dunning basis sets are often used in studying non-covalent interactions. Considering our systems are bigger, we firstly optimized these complexes with a small Dunning basis set aug-cc-pVDZ and the corresponding interaction energy is marked as Δ*E*_1_. Then we optimized these complexes with another type of basis set 6-311++G(d,p) and the corresponding interaction energy is represented as Δ*E*_2_. 

One can see from Table 1 that Δ*E*_1_ is −112.42 kJ/mol in **4**, which is much larger than −16.62 kJ/mol in **1**. On the other hand, the aug-cc-pVDZ basis set brings out a much shorter Si···O separation (2.0125 Å) in **4** than that in **1** (2.8562 Å). That is, the methyl substituent greatly enhances the tetrel bond. Obviously, this is not possible owing to the weak electron-donating ability of the methyl group. In addition, Δ*E*_1_ is almost equal in **6** and **7**, which is also inconsistent with the different electron-withdrawing ability of both –F and –CHO groups. More impossibly, the interaction energies in **6**, **7** and **9** are much larger at the MP2/aug-cc-pVDZ level than those at the MP2/aug’-cc-pVTZ level. This shows that the small Dunning aug-cc-pVDZ basis set is sometimes not reliable in studying the Si···O tetrel bonds and it is cautious to use this basis set in studying tetrel bonds since this basis set could provide some erroneous information in the structures and energies for tetrel bonded complexes.

For the 6-311++G(d,p) interaction energy, it is found that Δ*E*_2_ in **4** is smaller than that in **1**, consistent with the weak electron-donating nature of the methyl group. However, the –F substituent results in a larger Δ*E*_2_ than the –CHO one and an almost equal Δ*E*_2_ is caused by –NO_2_ and –SO_3_H substituents. This result disagrees with the relative magnitude of electron-withdrawing ability of these substituents. Accordingly, we think that 6-311++G(d,p) basis set is also unsuitable to study the Si···O tetrel bonds. To compensate the interaction energies of small 6-311++G(d,p) basis set in the above complexes, we calculated the interaction energies with the larger basis sets aug-cc-pVTZ and 6-311++G(3df,2p) on the MP2/6-311++G(d,p) geometries and the corresponding interaction energies are denoted as Δ*E*_3_ and Δ*E*_4_, respectively. As expected, the interaction energies with the larger basis sets are larger than those with the smaller ones. However, the interaction energies in the above complexes are not improved and even become almost equal in **6** and **7** with the two larger basis sets. Thus such correction for the interaction energy is not feasible.

It was demonstrated that MP2/aug-cc-pVTZ has often been used in studying tetrel bonded complexes [9,17,32,34]. However, this method is unpractical in large systems involving tetrel bonds owing to the computational cost and resource in our current condition. Based on the above results, these complexes were optimized at the MP2 level with aug’-cc-pVTZ basis set, in which the cc-pVTZ basis set is used for H atom. Importantly, MP2/aug’-cc-pVTZ method has successfully used to study many pnicogen bonded complexes [49,50,51]. Thus the following discussion is based on the MP2/aug’-cc-pVTZ results.

### 3.2. Interaction Energies and Geometries

The MP2/aug’-cc-pVTZ optimized structures are shown in Appendix A. Although these complexes are connected mainly by a tetrel bond, they show an observed difference in structures. For X = H, NH_2_, OCH_3_, CH_3_ and OH, the –SiF_3_ group attacks the oxygen atom of DMSO along the inner side of both methyl groups. A reverse conformation is adopted for X = F. The –SiF_3_ group is introduced into the oxygen atom of DMSO along the direction of one methyl group when X = CHO, CN, NO_2_, and SO_3_H. Even so, we think that other interactions are very weak relative to the tetrel bond and do not change the tendency in stability of these complexes.

The interaction energy is calculated to be −18.87 kJ/mol in PTS···DMSO (**1**). Thus the tetrel bond in **1** belongs to a moderate interaction, having a comparable strength with the hydrogen bond in H_2_O···H_2_O (about 20 kJ/mol at the MP2/CBS level [52]). This implies that the tetrel bond in **1** has an important effect on its structures and properties. The interaction energy in **1** is much smaller than that in SiF_3_X···DMSO (X =halogen), which is about 129–157 kJ/mol [32]. The interaction energy is −14.53 kJ/mol in PhSiH_3_···DMSO (Appendix A), which is smaller than that in **1**. This is attributed to the weak acidity of Si atom in PhSiH_3_, confirmed by the smaller σ-hole on the Si atom of PhSiH_3_. The dipole moment is 0.85 and 3.41 D for PhSiH_3_ and PhSiF_3_, respectively. Thus polarization is also responsible for the larger interaction energy in **1** (Appendix A). The interaction energy is −5.09 kJ/mol in PhCF_3_···DMSO, much smaller than that in **1**. The smaller σ-hole on the carbon atom and dipole moment (3.10 D) of PhCF_3_ lead to the weaker tetrel bond in PhCF_3_···DMSO. The interaction energy is ~3.34 kJ/mol in PhCF_3_···Cl^−^ at the CCSD(T)/def2-TZVP level of theory [53], which is close to that in PhCF_3_···DMSO. This implies that like anions, DMSO has a strong affinity for the tetrel atom.

The para H atom of PhSiF_3_ can be replaced by various substituents including electron-donating groups (NH_3_, OCH_3_, CH_3_, and OH) and electron-withdrawing groups (F, CHO, CN, NO_2_, and SO_3_H). The interaction energy climbs from a minimum of −16.29 kJ/mol to −122.51 kJ/mol for the strongest *p*-SO_3_H-PhSiF_3_ electron acceptor. As expected, the electron-donating groups weaken the tetrel bond and the electron-withdrawing groups enhance the tetrel bond. The weakening of tetrel bond varies in the order OH < CH_3_ < OCH_3_ < NH_3_, and the enhancing of tetrel bond is in the sequence F < CHO < CN < NO_2_ < SO_3_H. Moreover, the electron-donating groups have a small weakening effect on the strength of the tetrel bond, while the electron-withdrawing groups have a prominent enhancing effect on the strength of tetrel bond. For the electron-withdrawing groups, the interaction energy of the tetrel bond exceeds −95 kJ/mol, corresponding to a very strong tetrel bond. This influence of electron-withdrawing substituents is much greater than that in halogen bonded pyridine complexes [54]. Moreover, any electron-withdrawing substituent in the aromatic tetrel bond donors can cause the prominent enhancement of the tetrel bond. A simple F substitution could realize this target and the interaction energy has a slight increase when the electron-withdrawing ability of a group exceeds that of CN. Consequently, introducing an electron-withdrawing substituent in the aromatic tetrel bond donors is a very efficient method for strengthening tetrel bonds.

We have represented the Hammett’s plot for the tetrel bonded complexes studied here. In Figure 2, we have plotted the interaction energies versus the aromatic substituent constant (δ) for the benzene complexes. A good degree of correlation is obtained for the electron-donating and electron-withdrawing substituents, respectively. Thus it can be concluded that the aromatic substituent constant (δ) can be used to measure trends in tetrel bonding stability. Interestingly, the slope of the regression plot in the electron-withdrawing substituents is greatly larger than that in the electron-donating substituents, indicating that the influence of the electron-withdrawing substituents on the interaction energy is bigger than that of the electron-donating substituents. In addition, the different regression plots for both electron-donating and electron-withdrawing substituents imply that the influence of the substituents on the interaction energies is caused by induction effects and that resonance effects are also important.

To study the electrostatic nature of the tetrel bonding interaction in the aromatic complexes, we have computed the most positive MEP value of the σ-hole at the C–Si end in the tetrel bond donors. These values are shown in Appendix A. Also we have represented the MEP values versus the interaction energies of the complexes in Figure 3. We have obtained a good relationship between the interaction energies of the complexes and the MEP values for the electron-donating and electron-withdrawing substituents, respectively. This result clearly indicates that the changes observed in the interaction energies of the complexes due to the different substituents are primarily caused by electrostatic effects. The dominant role of electrostatic interaction is also confirmed by the energy decomposition results (Appendix A). Moreover, the slope of the regression plot in the electron-withdrawing substituents is greatly larger than that in the electron-donating substituents, which is similar to that of the relationship between the interaction energy and Hammett constant (Figure 2).

Table 2 presents the binding distance in the complexes. The binding distance comes down from a maximum of 3.00 Å for the –NH_2_ substituent to 2.02 Å for –SO_3_H. It is much shorter than the sum of van der Waals Radii of Si and O atoms (~3.6 Å) [55]. A good quadratic relationship is found between the interaction energy and the Si···O separation (Figure 4). However, a linear relationship is usually present in hydrogen bonds [56]. This difference is mainly caused by the steric hindrance of three F atoms in the tetrel bonds. Along with this interaction strengthening, one sees a concomitant contraction of the intermolecular separation. Moreover, the interaction energy is more sensitive to the binding distance for the strong tetrel bonds. Namely, the small change of the binding distance in the range of the short Si···O separation corresponds to the large variation of the interaction energy.

The formation of tetrel bonds results in an elongation of C–Si and S=O bonds (Table 2). The elongation of C–Si bond is greater than that of S=O bond. A good linear relationship is found between the elongation of C–Si bond and the interaction energy (Appendix A). This indicates that the magnitude of C–Si bond elongation can also be an indicator of tetrel bonding strength. The largest elongation of C–Si bond amounts to about 0.05 Å in **10**. Generally, the elongation of the S=O bond is also in close correspondence with the interaction energy (Appendix A). The formation of the tetrel bond causes a deformation of the –SiF_3_ group, which can be estimated with the change of angle C–Si–F (Δα) in the complex relative to the monomer. Δα is negative, indicating that there is a tendency to form a pentacoordinate silicon complex. The larger Δα corresponds to a stronger tetrel bond (Appendix A).

The interaction energy increases by −100 kJ/mol due to the –CN substituent in the PhSiF_3_ complex. However, this substituent strengthens the tetrel bond by only −5.73 kJ/mol in the PhSiH_3_ complex (Appendix A). The –CN substituent makes an increase of 37.07 and 35.32 kJ/mol for the MEP of the σ-hole on the Si atom in PhSiF_3_ and PhSiH_3_, respectively. On the other hand, when the complex varies from PhSiF_3_/PhSiH_3_ to its –CN substituents, Δα is increased by about 10^o^ in the case of PhSiF_3_ but is almost not changed in the case of PhSiH_3_. This indicates that the deformation energy is also important in influencing the interaction energy of the tetrel bond by the electron-withdrawing substituent.

### 3.3. NBO Analyses

It has been demonstrated that charge transfer from the lone pair of a Lewis base into the T–X anti-bonding orbital also stabilizes the tetrel bonded complexes of TH_3_X [7]. The similar orbital interaction of Lp_O_→σ*_C–Si_ is also present in the Si···O tetrel bond of *p*-X-PhSiF_3_···DMSO complex. Interestingly, another important orbital interaction of Lp_O_→*p**_Si_ is found in the Si···O tetrel bond. According to the corresponding second-order perturbation energy in Table 3, one can see that the latter orbital interaction is far stronger than the former one in the Si···O tetrel bond. That is, the Lp_O_→*p**_Si_ orbital interaction plays a more important role in stabilizing the strong Si···O tetrel bond than Lp_O_→σ*_C–Si_. This is different from what is found in hydrogen bonding, where the lone pair charge is mainly transferred into the H–X anti-bonding orbital [57], and in weak tetrel bonding, where the Lp_O_→σ*_T–X_ orbital interaction is dominant [7]. The presence of Lp_O_→*p**_Si_ orbital interaction can be taken as a predictor of a strong tetrel bond with the formation of a partially covalent bond [32] and it is also used to explain the deformation of –SiF_3_ group in the complexes. The charge transfer from the lone pair of the oxygen atom to the C–Si anti-bonding orbital is also reflected in the C–Si bond elongation after complex formation although this charge transfer is small. The second-order perturbation energy due to the Lp_O_→*p**_Si_ orbital interaction exhibits a good linear relationship with the interaction energy (Figure 5). The former is much larger in magnitude than the latter for the electron-withdrawing substituents. This result indicates that orbital interaction makes great contribution to the stabilization of strong tetrel bonded complexes. However, in the complexes with an electron-withdrawing substituent, the shorter Si···O separation brings out a big repulsive interaction (Appendix A), which could partly cancel the contribution of orbital interaction.

The charge transfer (Δ*q*) from the DMSO fragment to the *p*-X-PhSiF_3_ fragment is also given in Table 3. Δ*q* < 0 shows that charge is transferred from the DMSO fragment to the *p*-X-PhSiF_3_ fragment in the tetrel bond. Δ*q* is small (<0.01 e) in the complexes with an electron-donating substituent but is very large (>0.12 e) in the complexes with an electron-withdrawing substituent. The small Δ*q* supports the conclusion that electrostatic interaction is dominant in the weak Si···O interaction and the large Δ*q* is consistent with the fact that the strong Si···O tetrel bond has a nature of a partially covalent interaction.

There are also internal rearrangements within each monomer besides shifts from one molecule to another. The total electron density redistribution is displayed in Figure 6 for four representative complexes, where purple regions represent charge buildup and depletion of density is indicated by blue. There is a common pattern in all of these complexes. Firstly, a small increase occurs on the oxygen lone pair of DMSO, accompanied with a larger area of charge loss immediately to the left of the Si atom. Buildup is also observed on the three F atoms of –SiF_3_ group. The C–Si bond suffers a substantial loss of charge as indicated by the large blue area and this area is enlarged from the –NH_2_ to the –CN substituent.

### 3.4. AIM Analyses

In the above analyses, we have pointed out that there exist other interactions between DMSO and *p*-X-PhSiF_3_ and the strong tetrel bond in the complexes with an electron-withdrawing substituent exhibits a nature of partially covalent interaction. To confirm these, we performed an AIM analysis for these systems. Figure 7 shows the AIM bonding diagrams of three representative complexes. Interestingly, the tetrel bond is characterized by three F···O bond critical points (BCPs), not by a Si···O BCP. The similar result was also reported in F_4_Si···NCH complex [5]. Actually, there are some controversies and discussions on the physical meaning of the bond path term [58], although the bond paths show stabilizing and preferable interactions for the system being in the energetic minima [59]. Three CH···F BCPs are also found in **1**, confirming the presence of CH···F interactions. The H···F separation (~2.5 Å) in **1** is a little shorter than the sum of van der Waals Radii of H and F atoms (~2.67 Å) [55]. The interaction energy of CH···F interaction is about −2 kJ/mol between two neutral molecules [60]. Thus we think that the CH···F interactions in **1** are weak and they have an insignificant effect on the strength of the tetrel bond although they play a role in maintaining the conformation of complex. The paths of the BCPs and the nature of Si···O tetrel bond in the complexes with the electron-donating group (**2**–**5**) are the same as those in **1**. The Si···O tetrel bond in **6** is characterized with the presence of a Si···O BCP and no other BCPs are found in **6**. The Si···O BCP has the larger electron density and Laplacian in **6** than the F···O BCP in **1**. On the other hand, the energy density at the Si···O BCP is negative in **6**, indicating the partially covalent character of the Si···O tetrel bond [61]. The Si···O tetrel bond in **7** has the same Si···O BCP with the positive Laplacian and the negative energy density as that in **6**. Nevertheless, the electron density at the Si···O tetrel bond in **7** is a little larger than that in **6**, consistent with the strength of tetrel bond in both complexes. In addition, **7** has three similar intermolecular BCPs like **1**. In **9**, there is a F···C BCP besides the Si···O BCP. The former BCP is also taken as a predictor of a F···C tetrel bond. Similarly, the Si···O tetrel bond in **9** has the negative energy density but the larger electron density, corresponding to a stronger Si···O tetrel bond with a covalent character. The same paths of Si···O and F···C BCP are also found in **8** and **10**.

### 3.5. Competition

In the complexes of DMSO and *p*-X-PhSiF_3_ (X = NH_2_, OH, CHO, and NO_2_), there exist other interactions besides a tetrel bond. As expected, the proton of both NH_2_ and OH groups can form a hydrogen bond with the oxygen atom of DMSO. It is seen from Appendix A (Appendix A) that there is a π-hole on the carbon atom of –CHO and the nitrogen atom of –NO_2_, respectively. The π-hole is a region with positive MEPs that is perpendicular to an adjacent portion of the molecular framework [62]. Thus the oxygen atom of DMSO participates in the formation of both a π-hole tetrel bond with the carbon atom of –CHO and a π-hole pnicogen bond with the nitrogen atom of –NO_2_. The latter interaction has been evidenced in many complexes involving nitro compounds [63,64,65]. We have studied some π-hole tetrel bonds in the complexes of F_2_CX (X = Se and Te) [17,66]. However, no evidence is found for the π-hole tetrel bonds involving a formyl group. Figure 8 shows the AIM bonding diagrams of these complexes.

In *p*-SiF_3_-PhOH···DMSO (**11**), the conformation of the complex is similar to that of the most stable complex of DMSO and H_2_O [67], in which an OH···O hydrogen bond coexists with two CH···O interactions. Both OH···O hydrogen bond and CH···O interactions in **11** are characterized by an OH···O BCP and a CH···O BCP, respectively. Additionally, a CH···O BCP is also found between the C-H proton of the benzene ring and the oxygen atom of DMSO in **11**. The energy density at the OH···O BCP in **11** is negative, indicating that this hydrogen bond has a partially covalent character [61]. Thus the OH···O hydrogen bond in **11** is stronger than that in the complex of DMSO and H_2_O, consistent with the acidity of hydroxyl proton in both molecules of *p*-HO-PhSiF_3_ and H_2_O. The interaction energy is calculated to be −62.02 kJ/mol in **11**, which is much larger than that in **5**. As a result, the oxygen atom of DMSO is favorable to bind with the hydroxyl proton of *p*-HO-PhSiF_3_ than with the –SiF_3_ group.

In *p*-SiF_3_-PhNH_2_···DMSO (**12**), the conformation of the complex is different from that in **11** since the nitrogen atom of –NH_2_ cannot again bind with two C–H protons of DMSO simultaneously. The NH···O hydrogen bond is characterized by a NH···O BCP. Interestingly, both a C···S BCP and a CH···C BCP are present, where the carbon atom is adjoined to the –NH_2_ group in the benzene ring. The energy density at the NH···O BCP is positive, implying an electrostatic domination in this interaction. The interaction energy in **12** is larger than that in **2**, thus DMSO prefers to form a hydrogen bonded complex than a tetrel bonded complex with *p*-H_2_N-PhSiF_3_. The electron density at the H···O BCP shows that the NH···O hydrogen bond in **12** is weaker than the OH···O one in **11**, confirming the fact that the proton of –OH has a larger acidity than that of –NH_2_ [68].

In *p*-SiF_3_-PhCHO···DMSO (**13**), there is a C···O BCP between the carbon atom of –CHO and the oxygen atom of DMSO, corresponding to a π-hole tetrel bond. A CH···O interaction is present between the hydrogen atom of DMSO and the oxygen atom of –CHO, evidenced by a CH···O BCP. Its interaction energy is estimated to be −4 kJ/mol according to the formulas of Δ*E* = 1/2 × V, where V is the potential energy density at the CH···O BCP [69]. In addition, two CH···C BCPs are found between the hydrogen atom of DMSO and the carbon atom of benzene ring. Even so, we think that the π-hole tetrel bond is a dominant interaction in **13**. Of course, other weak interactions are also important in maintaining the conformation of the complex. The interaction energy in **13** amounts to about a third of that of **7**, indicating that the σ-hole tetrel bonded complex is dominant over the π-hole tetrel bonded complex.

In *p*-SiF_3_-PhNO_2_···DMSO (**14**), the π-hole pnicogen bond is characterized with the presence of a C···O BCP, where the carbon atom is connected with –NO_2_. Although the π-hole appears on the nitrogen atom of –NO_2_, no bond path is found between the nitrogen atom of –NO_2_ and the oxygen atom of DMSO. This is different from the π-hole pnicogen bonded complexes between nitryl derivatives (NO_2_X, X = CN, F, Cl, Br, NO_2_, OH, CCH, and C_2_H_3_) and molecules acting as Lewis bases (H_2_O, H_3_N, CO, HCN, HNC and HCCH), where a BCP is associated with a bond path between the nitrogen atom of the nitryl derivative and one of the atoms of the electron donor molecule [65]. Besides, three CH···O BCPs are found between the hydrogen atom of DMSO and the oxygen atom of –NO_2_. The interaction energy is −35.83 kJ/mol in **14**, which is less than a third of that in **9**. This means that the σ-hole tetrel bonded complex is more stable than the π-hole pnicogen bonded complex. The interaction energy of π-hole pnicogen bond varied from −4.7 kJ/mol in NO_2_OH···CO to −22.1 kJ/mol in NO_2_NO_2_···NH_3_ [65]. We think that the interaction energy of π-hole pnicogen bond in **14** is also within this range, and this conclusion is confirmed by the small electron density and the positive energy density at the C···O BCP.

To unveil the role of the –SiF_3_ group in the above interactions, we also studied the complexes of DMSO and PhX (X = NH_2_, OH, CHO, NO_2_) by replacing this group with a hydrogen atom. As expected, their structures are similar to those of *p*-SiF_3_-PhX···DMSO. The interaction energies are −42.32, −56.89, −32.03, and −31.69 kJ/mol for X = NH_2_, OH, CHO, and NO_2_, respectively. Obviously, they are smaller than those in the corresponding complexes of *p*-SiF_3_-PhX. Accordingly, the –SiF_3_ group in the benzene ring plays an electron-withdrawing role in the above interactions.

## 4. Conclusions

The complexes of DMSO and *p*-X-PhSiF_3_ (X = H, NH_3_, OCH_3_, CH_3_, OH, F, CHO, CN, NO_2_, and SO_3_H) have been investigated in views of the structures, energies, NBO, and AIM. The following conclusions have been reached.

The basis set has an important influence on the structures and interaction energies of Si···O tetrel bonded complexes. It is necessary to use sufficiently flexible basis sets, such as aug’-cc-pVTZ, to get reliable results for the Si···O tetrel bonds.

The interaction energy of the Si···O tetrel bond varies in a wide range from the –NH_2_ (−16.29 kJ/mol) to the –SO_3_H group (−122.51 kJ/mol). The effect of the substituents in the tetrel bonded complexes follows the expected trend. Namely, this complex is favoured when an electron-withdrawing group occurs in the tetrel donor but an opposite result is obtained for the electron-donating substituent. Moreover, the weakening effect of the electron-donating substituent is trifling, while the enhancing impact of the electron-withdrawing group is wondrously prominent.

The substituents can also affect the nature of the Si···O tetrel bond. The Si···O tetrel bond is dominated by the electrostatic interaction in the complexes with the electron-donating substituent, while it has some degree of covalent character in the complexes with the electron-withdrawing group.

A strong linear relationship is established between the interaction energy and the Hammett’s constant for the electron-donating and electron-withdrawing substituents, respectively. The MEP value on the σ-hole of the Si atom in *p*-X-PhSiF_3_ strongly correlates with the interaction energies of the complexes for the electron-donating and electron-withdrawing substituents, respectively. This implies that deformation energy mainly caused by the formation of the partially covalent interaction is crucial in the strong Si···O tetrel bonds.

There is competition between the Si···O tetrel bond and another interaction in the complexes of DMSO and *p*-X-PhSiF_3_ (X = NH_2_, OH, CHO, and NO_2_). The hydrogen bonded complexes are more stable than the Si···O tetrel bonded ones for X = NH_2_ and OH, while the σ-hole tetrel bond prefers the π-hole interaction for X = CHO and NO_2_.

## Figures and Tables

**Figure 1 molecules-26-07231-f001:**
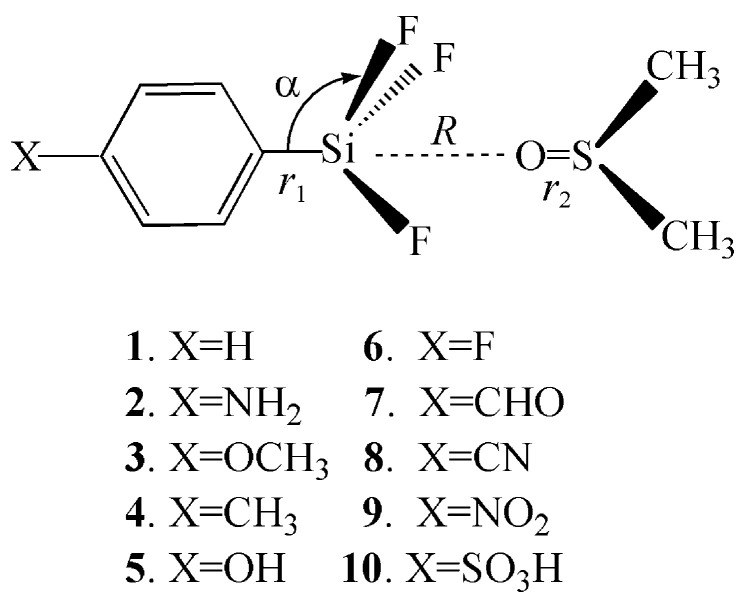
Scheme of tetrel bonded complexes.

**Figure 2 molecules-26-07231-f002:**
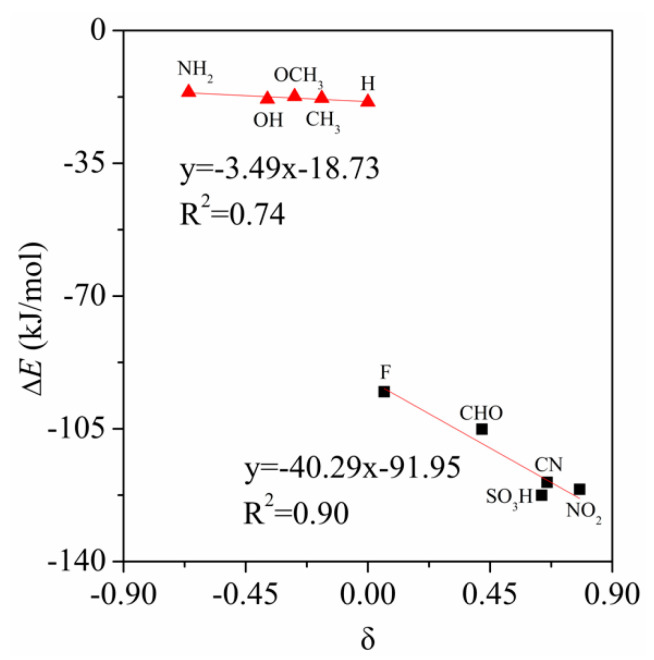
Hammett’s plot of the tetrel bonded complexes.

**Figure 3 molecules-26-07231-f003:**
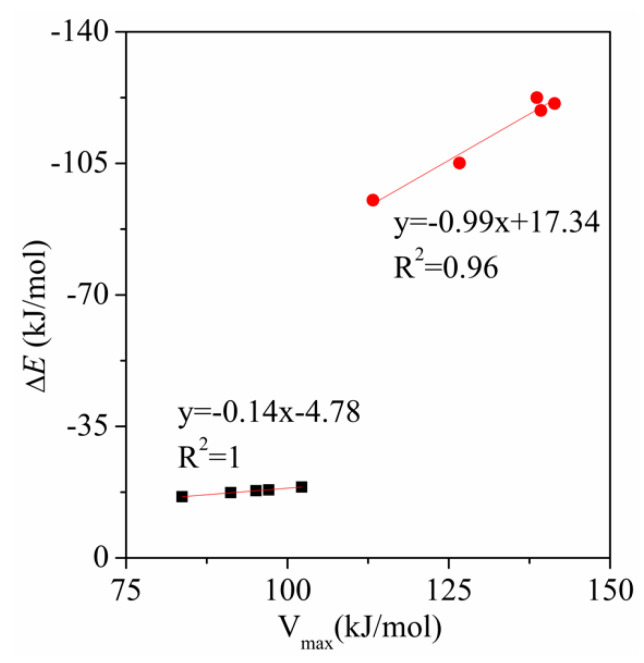
Regression plots of the interactions energy (Δ*E*) *versus* MEP value.

**Figure 4 molecules-26-07231-f004:**
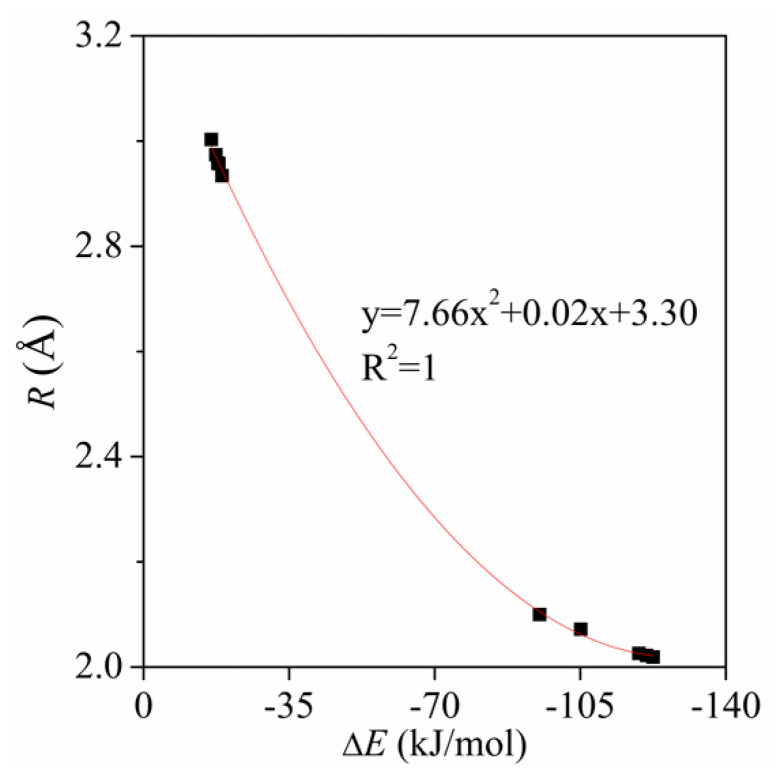
Regression plots of the interactions energy (Δ*E*) *versus* binding distance (*R*).

**Figure 5 molecules-26-07231-f005:**
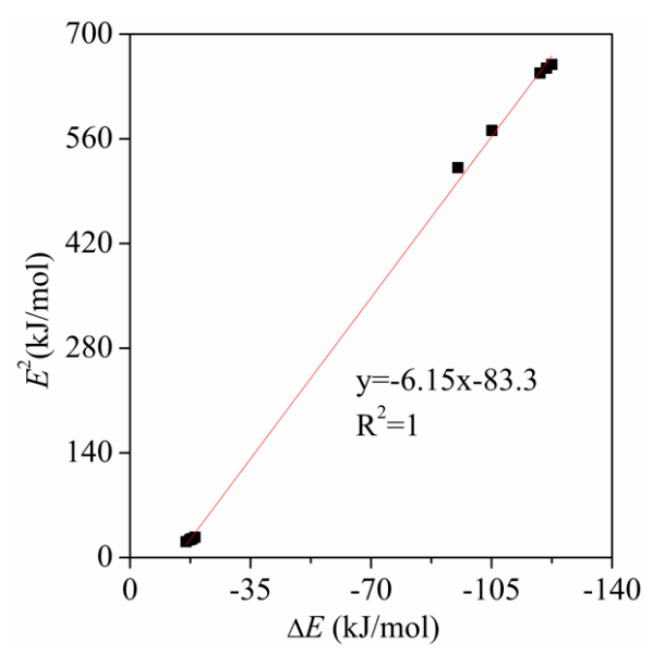
Regression plots of the interactions energy (Δ*E*) *versus* second-order perturbation energy (*E*^2^) due to the Lp_O_→*p**_Si_ orbital interaction.

**Figure 6 molecules-26-07231-f006:**
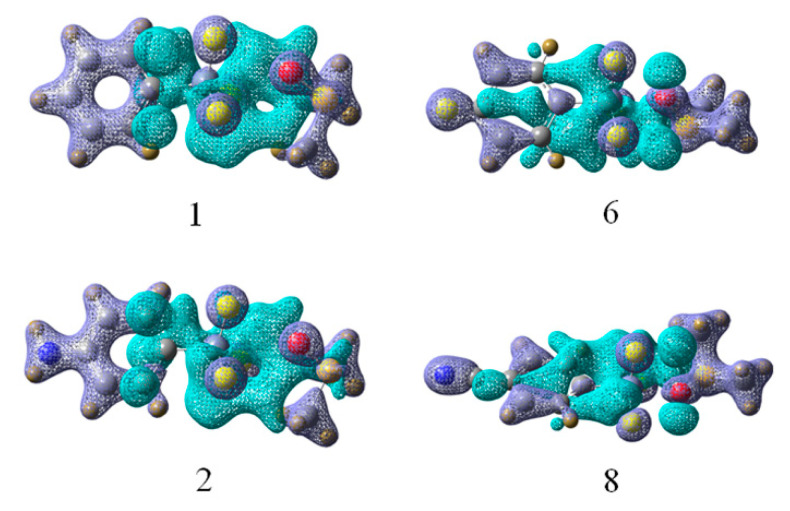
Density shifts occurring in the four tetrel bonded complexes of **1**, **2**, **6**, and **8** upon formation of each complex. Purple regions indicate density increase, blue a decrease. Contours are shown at the 0.1 au level.

**Figure 7 molecules-26-07231-f007:**
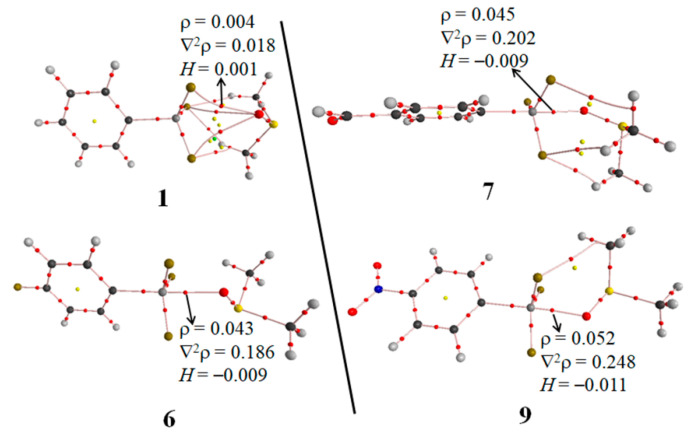
Molecular graph of the four tetrel bonded complexes of **1**, **6**, **7**, and **9**.

**Figure 8 molecules-26-07231-f008:**
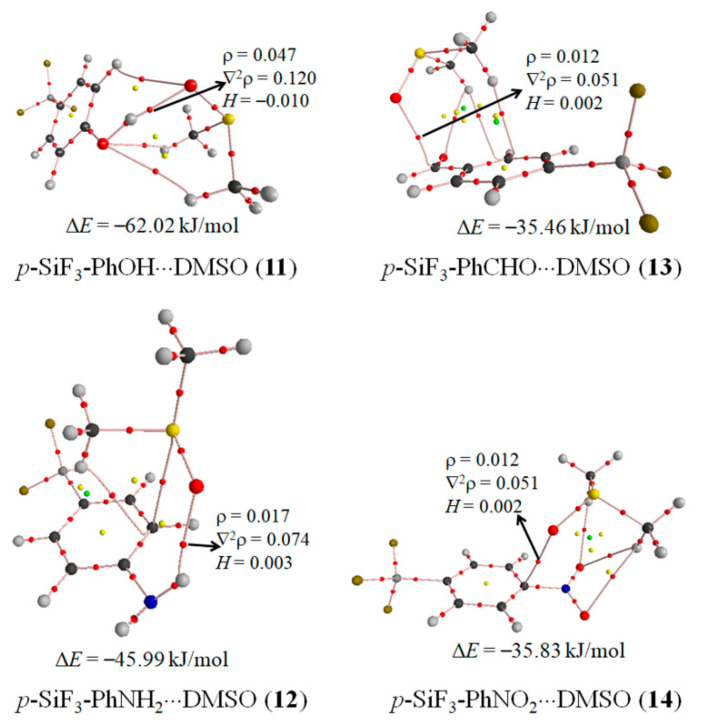
Molecular graph of the complexes between DMSO and *p*-SiF_3_-PhX (X = NH_2_, OH, CHO, and NO_2_).

**Table 1 molecules-26-07231-t001:** Interactions energies (Δ*E*, kJ/mol) at the levels of MP2/aug-cc-pVDZ//MP2/aug-cc-pVDZ (1), MP2/6-311++G(d,p)//MP2/6-311++G(d,p) (2), MP2/aug-cc-pVTZ//MP2/6-311++G(d,p) (3), MP2/6-311++G(3df,2p)// MP2/6-311++G(d,p) (4), and MP2/aug’-cc-pVTZ//MP2/aug’-cc-pVTZ (5) in the ten complexes of *p*-X-PhSiF_3_···DMSO.

	X=	Δ*E*_1_	Δ*E*_2_	Δ*E*_3_	Δ*E*_4_	Δ*E*_5_
**1**	H	−16.62	−15.41	−22.07	−19.58	−18.87
**2**	NH_2_	−16.45	−12.26	−18.76	−16.43	−16.29
**3**	OCH_3_	−13.96	−13.15	−19.69	−17.31	−17.38
**4**	CH_3_	−112.42	−13.06	−19.66	−17.31	−17.89
**5**	OH	−20.95	−14.57	−21.17	−18.73	−18.10
**6**	F	−136.22	−102.08	−112.33	−107.64	−95.23
**7**	CHO	−136.72	−95.25	−112.48	−107.42	−105.10
**8**	CN	non	−116.59	−135.46	−130.46	−119.10
**9**	NO_2_	−152.15	−119.47	−138.19	−133.21	−120.95
**10**	SO_3_H	non	−119.00	−137.67	−132.56	−122.51

**Table 2 molecules-26-07231-t002:** Binding distance (*R*, Å), C–Si bond length (*r*_1_, Å), change of C–Si (1) and S=O (2) bond lengths (Δ*r*, Å), and change of angle C–Si–F (Δα, degree) in the complexes of *p*-X-PhSiF_3_···DMSO.

	X=	*R*	*r* _1_	Δ*r*_1_	Δ*r*_2_	Δα
**1**	H	2.9336	1.8453	0.0095	0.0035	−3.0
**2**	NH_2_	3.0032	1.8372	0.0088	0.0028	−2.6
**3**	OCH_3_	2.9740	1.8395	0.0090	0.0031	−2.7
**4**	CH_3_	2.9581	1.8429	0.0092	0.0032	−2.8
**5**	OH	2.9566	1.8405	0.0093	0.0033	−2.8
**6**	F	2.0996	1.8790	0.0439	0.0333	−12.3
**7**	CHO	2.0715	1.8861	0.0460	0.0268	−12.8
**8**	CN	2.0265	1.8910	0.0504	0.0384	−13.4
**9**	NO_2_	2.0221	1.8915	0.0501	0.0388	−13.4
**10**	SO_3_H	2.0188	1.8926	0.0506	0.0392	−13.5

**Table 3 molecules-26-07231-t003:** Second-order perturbation energies (*E*^2^, kJ/mol) due to the Lp_O_→*p**_Si_ (1) and Lp_O_→σ*_C-Si_ (2) orbital interactions and charge transferred (Δ*q*, e) from DMSO to *p*-X-PhSiF_3_ fragment in the complexes of *p*-X-PhSiF_3_···DMSO.

	X=	*E*^2^(1)	*E*^2^(2)	Δ*q*
**1**	H	23.70	2.76	−0.0100
**2**	NH_2_	18.85	2.30	−0.0078
**3**	OCH_3_	20.69	2.47	−0.0086
**4**	CH_3_	21.82	2.55	−0.0092
**5**	OH	21.94	2.55	−0.0092
**6**	F	469.04	20.06	−0.1233
**7**	CHO	483.83	22.40	−0.1329
**8**	CN	583.90	23.83	−0.1440
**9**	NO_2_	590.55	24.16	−0.1455
**10**	SO_3_H	595.44	24.20	−0.1465

## Data Availability

Not applicable.

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
