# Peer review of "Tetrel Bonds between Phenyltrifluorosilane and Dimethyl Sulfoxide: Influence of Basis Sets, Substitution and Competition"

_molecules, 2021, doi:10.3390/molecules26237231_

Round 1

Reviewer 1 Report

This manuscript could be accepted for publication in Molecules. The novelty of presented research (the phenomenon of tetrel bonds between phenyltrifluorosilane and dimethyl sulfoxide was studied theoretically) and it's importance for the scientific community of supramolecular chemistry are high. The introduction provide sufficient background. The research methodology is adequate. The results are clearly presented. The conclusions supported by the data. The manuscript good illustrated and interesting to read. English language and style are fine. I have only couple of minor suggestions. Firstly, I suggests to cite some relevant papers in 2. Theoretical Methods section to show that other researchers also used similar methodology for theoretical calculations of supramolecular organization via different noncovalent interactions: hydrogen bonds (Phys. Chem. Chem. Phys. 2016, V. 18. P. 14104.), halogen bonds (Cryst. Growth Des. 2017, V. 17. P. 1353.), chalcogen bonds (CrystEngComm 2021, V. 23. P. 4607.), π-stacking (J. Mol. Struct. 2016, V. 1104. P. 19.; CrystEngComm 2021, V. 23. P. 6409.). Secondly, it would be useful for potential readers if authors add xyz-files or a table with Cartesian atomic coordinates for all optimized equilibrium model strictures.
Overall, this manuscript could be accepted for publication after minor revisions.

Author Response

Response for the comments of reviewer 1:

This manuscript could be accepted for publication in Molecules. The novelty of presented research (the phenomenon of tetrel bonds between phenyltrifluorosilane and dimethyl sulfoxide was studied theoretically) and it's importance for the scientific community of supramolecular chemistry are high. The introduction provides sufficient background. The research methodology is adequate. The results are clearly presented. The conclusions supported by the data. The manuscript is good illustrated and interesting to read. English language and style are fine. I have only couple of minor suggestions.

Comment 1: Firstly, I suggest to cite some relevant papers in 2. Theoretical Methods section to show that other researchers also used similar methodology for theoretical calculations of supramolecular organization via different noncovalent interactions: hydrogen bonds (Phys. Chem. Chem. Phys. 2016, V. 18. P. 14104.), halogen bonds (Cryst. Growth Des. 2017, V. 17. P. 1353.), chalcogen bonds (CrystEngComm 2021, V. 23. P. 4607.), π-stacking (J. Mol. Struct. 2016, V. 1104. P. 19.; CrystEngComm 2021, V. 23. P. 6409.).

Response: These references have been cited. The following sentence has been added: The similar theoretical methodology was also performed for other types of noncovalent interactions [40-44].

[40] Serebryanskaya, T. V.; Novikov, A. S.; Gushchin, P. V.; Haukka, M.; Asfin, R. E.; Tolstoy, P. M.; Kukushkin, V. Y. Identification and H(D)-bond energies of C–H(D)⋯Cl interactions in chloride–haloalkane clusters: A combined X-ray crystallographic, spectroscopic, and theoretical study. Phys. Chem. Chem. Phys. 2016, 18, 14104-14112.

[41] Ivanov, D. M.; Kinzhalov, M. A.; Novikov, A. S.; Ananyev, I. V.; Romanova, A. A.; Boyarskiy, V. P.; Haukka, M.; Kukushkin, V. Y. H2C(X)–X···X (X = Cl, Br) halogen bonding of dihalomethanes. Cryst. Growth Des. 2017, 17, 1353-1362.

[42] Novikov, A.S.; Gushchin, A. Trinuclear molybdenum clusters with sulfide bridges as potential anionic receptors via chalcogen bonding. CrystEngComm 2021, 23, 4607-4614.

[43] Ivanov, D. M.; Kirina, Y. V.; Novikov, A. S.; Starova, G. L.; Kukushkin, V. Y. Efficient π-stacking with benzene provides 2D assembly of trans-[PtCl2(p-CF3C6H4CN)2]. J. Mol. Struct. 2016, 1104, 19-23.

[44] Abramov, P.A.; Novikov, A.S.; Sokolov, M.N. Interactions of aromatic rings in the crystal structures of hybrid polyoxometalates and Ru clusters. CrystEngComm 2021, 23, 6409-6417.

Reviewer 2 Report

In this manuscript, An et al. present their theoretical studies of tetrel bond between differently substituted phenyltrifluorosilane and DMSO. This work continues the series of articles presented by the same group (see, for example, 10.1039/C7RA02068F), focusing on  ab inition calculations of tetrel bonds. The main idea of this report is related to the proper choice of basis sets in order to achieve reliable results.

The calculations are performed in due order, and their results do not contradict the "chemical common sense" - in other words, most likely, the main conclusion on basis sets seems to be correct. It is unlikely that this work will attract too big attention - it is rather specialized. However, its quality is sufficient for publication in its present form.

Author Response

Response for the comments of reviewer 2:

In this manuscript, An et al. present their theoretical studies of tetrel bond between differently substituted phenyltrifluorosilane and DMSO. This work continues the series of articles presented by the same group (see, for example, 10.1039/C7RA02068F), focusing on ab inition calculations of tetrel bonds. The main idea of this report is related to the proper choice of basis sets in order to achieve reliable results. The calculations are performed in due order, and their results do not contradict the "chemical common sense" - in other words, most likely, the main conclusion on basis sets seems to be correct. It is unlikely that this work will attract too big attention - it is rather specialized. However, its quality is sufficient for publication in its present form.

Response: Thanks for the appraisal for our paper.

Reviewer 3 Report

In my opinion, the presented manuscript presents interesting and reliable results and can be published almost in present form.
I think that sample drawings for a selected molecule or two from supplementary materials should be included in the main text of the publication. I mean Fig. 2 and Fig. 3. e.g. showing the attack of the -SiF3 group on the oxygen atom from different sides or the MEP for systems with additional interactions, i.e. H-bonds.

One negative note - there is no graphical abstract.

Author Response

Response for the comments of reviewer 3:

In my opinion, the presented manuscript presents interesting and reliable results and can be published almost in present form.

Comment 1: I think that sample drawings for a selected molecule or two from supplementary materials should be included in the main text of the publication. I mean Fig. 2 and Fig. 3. e.g. showing the attack of the -SiF3 group on the oxygen atom from different sides or the MEP for systems with additional interactions, i.e. H-bonds.

Response: There are eight figures in the text and figure 1 shows the scheme of tetrel bonded complexes, thus we do not provide some drawings for the optimized structures from the supplementary materials to the text.

Yes, the oxygen atom of DMSO can attack the -SiF3 group from different sides since there are four σ-holes on the Si atom, however, we focused on the only one along the C-Si bond since such contact may avoid other interactions. Even so, other weak interactions are also present, which has been pointed out in the crude manuscript (Even so, we think that other interactions are much weak relative to the tetrel bond and do not change the tendency in stability of these complexes.)

Comment 2: One negative note - there is no graphical abstract.

Response: This has been added.